# The Quantum Memory Matrix: A Unified Framework for the Black Hole Information Paradox

**DOI:** 10.3390/e26121039

**Published:** 2024-11-30

**Authors:** Florian Neukart, Reuben Brasher, Eike Marx

**Affiliations:** 1Leiden Institute of Advanced Computer Science, Leiden University, Gorlaeus Gebouw-BE-Vleugel, Einsteinweg 55, 2333 Leiden, The Netherlands; 2Terra Quantum AG, Kornhausstrasse 25, 9000 St. Gallen, Switzerland; rb@terraquantum.swiss (R.B.); eike@terraquantum.swiss (E.M.)

**Keywords:** quantum mechanics, general relativity, black hole information paradox, quantum information, space–time quantization, quantum gravity, Hawking radiation, unitarity, quantum imprints, information retrieval, cosmology, quantum field theory, holographic principle, loop quantum gravity, gravitational waves

## Abstract

We present the Quantum Memory Matrix (QMM) hypothesis, which addresses the longstanding Black Hole Information Paradox rooted in the apparent conflict between Quantum Mechanics (QM) and General Relativity (GR). This paradox raises the question of how information is preserved during black hole formation and evaporation, given that Hawking radiation appears to result in information loss, challenging unitarity in quantum mechanics. The QMM hypothesis proposes that space–time itself acts as a dynamic quantum information reservoir, with quantum imprints encoding information about quantum states and interactions directly into the fabric of space–time at the Planck scale. By defining a quantized model of space–time and mechanisms for information encoding and retrieval, QMM aims to conserve information in a manner consistent with unitarity during black hole processes. We develop a mathematical framework that includes space–time quantization, definitions of quantum imprints, and interactions that modify quantum state evolution within this structure. Explicit expressions for the interaction Hamiltonians are provided, demonstrating unitarity preservation in the combined system of quantum fields and the QMM. This hypothesis is compared with existing theories, including the holographic principle, black hole complementarity, and loop quantum gravity, noting its distinctions and examining its limitations. Finally, we discuss observable implications of QMM, suggesting pathways for experimental evaluation, such as potential deviations from thermality in Hawking radiation and their effects on gravitational wave signals. The QMM hypothesis aims to provide a pathway towards resolving the Black Hole Information Paradox while contributing to broader discussions in quantum gravity and cosmology.

## 1. Introduction

The unification of Quantum Mechanics (QM) and General Relativity (GR) remains one of the most profound challenges in modern physics. Quantum mechanics governs the microworld of particles and fields, emphasizing probabilistic behavior, superposition, and entanglement, and describes unitary evolution by the Schrödinger equation. In contrast, GR provides a geometric description of gravity as the curvature of space–time determined by the Einstein field equations, and excels in describing large-scale structures and high-energy cosmic phenomena. Despite their individual successes, QM and GR are fundamentally incompatible in domains where both quantum effects and intense gravitational fields are significant, such as near singularities or black holes. Developing a theory that consistently incorporates both QM and GR is essential for progressing toward a unified framework of quantum gravity and deepening our understanding of the universe.

The Black Hole Information Paradox exemplifies this discord between QM and GR [1]. Classically, black holes are regions from which nothing can escape after crossing the event horizon, not even light [2,3]. Black holes appear to be defined solely by their mass, charge, and angular momentum, with no trace of the information contained in the matter that formed them, as stated in the no-hair theorem [4]. However, Stephen Hawking’s discovery of Hawking radiation [5] arising from quantum effects near the event horizon implies that black holes can emit radiation, and ultimately even evaporate. This radiation, characterized by a blackbody spectrum at the Hawking temperature
(1)$$T_{H}=\frac{\hbar c^{3}}{8\pi GMk_{B}},$$
is thermal and carries no apparent information about the black hole’s contents, suggesting irreversible loss of information with the disappearance of such a black hole [1,5]. This apparent violation of quantum mechanical unitarity raises fundamental questions about information conservation, one of the cornerstones of QM [6,7]. Addressing this paradox requires a new approach capable of reconciling QM and GR by preserving unitarity and information conservation while respecting the equivalence principle and maintaining compatibility with observed four-dimensional space–time [8,9,10]. An ideal solution would not only be theoretically sound but also offer an experimentally testable mechanism for information retrieval [11]. The Quantum Memory Matrix (QMM) hypothesis proposed herein introduces a new framework that reconceptualizes space–time itself as a dynamic quantum information reservoir. Unlike existing theories that rely on holographic mappings [9,12], nonlocal interactions [13], or topological constructs [14], the QMM hypothesis posits that information from quantum interactions is stored directly within the space–time structure through quantum imprints at the Planck scale. This approach offers several distinctive advantages:**Local Information Encoding**: Traditional models such as the holographic principle [8,9] and AdS/CFT correspondence [12] transfer information to lower-dimensional boundaries. These models rely on specific geometries such as Anti-de Sitter (AdS) space, which do not match our observed universe’s de Sitter geometry [15]. In contrast, the QMM encodes information locally within quantized units of space–time, maintaining information conservation within the bulk rather than relying on external boundaries.**Preservation of the Equivalence Principle and Smoothness at the Event Horizon**: Models such as firewall [16] and remnant theories [17] challenge the smoothness of space–time near the event horizon, violating the equivalence principle [18]. The QMM embeds information storage in the local granular structure of space–time, preserving smoothness and allowing infalling observers to experience an unperturbed horizon crossing, as predicted by GR.**Compatibility with Quantum Mechanical Locality and Causality**: Operating through local interactions that respect causality, QMM avoids nonlocal mechanisms where information might “leak” from a black hole [13]. Instead, information is stored within the fabric of space–time itself, preserving the causal structure and offering coherence between QM and GR.**Intrinsic Mechanism for Information Retrieval**: Unlike speculative models that lack a retrieval process, QMM’s quantum imprints evolve with the quantum states they encode, allowing a retrievable and dynamic record of black hole information without requiring new dimensions, wormholes, or additional symmetries [19,20].**Independence from Specific Geometries or Exotic Matter**: Many existing theories depend on specific geometries or hypothetical forms of matter, such as extra dimensions [21]; however, the QMM hypothesis is compatible with our four-dimensional space–time and free of such speculative constructs, enhancing its potential for experimental validation.

The QMM hypothesis, operating within familiar four-dimensional space–time and aligning with both QM and GR, may offer an empirically accessible framework for tackling the Black Hole Information Paradox. In the following sections, we present the theoretical foundations and comprehensive mathematical structure of the QMM, detailing how quantum imprints function and how this framework addresses information conservation in black hole dynamics.

## 2. The Quantum Memory Matrix Hypothesis

### 2.1. Fundamental Principles

The QMM hypothesis proposes a distinct framework wherein the fabric of space–time functions as an active quantum information reservoir. This paradigm fundamentally shifts space–time from a passive setting to an interactive entity capable of storing and transferring information. Below, we elaborate on the foundational principles of the QMM hypothesis, which seeks to provide a concrete mechanism for the preservation and retrieval of information in quantum gravitational contexts such as black holes. The QMM hypothesis introduces a distinct interpretation of space–time’s role in quantum processes.

#### 2.1.1. Quantization of Space–Time

At the Planck scale ($l_{P}\approx 1.616\times 10^{-35}$ m), space–time is proposed to be discretized into fundamental units, termed space–time quanta or quantum cells. Each quantum cell occupies a finite region and is associated with a finite-dimensional Hilbert space $H_{x}$, where *x* denotes its position in space–time. This discrete structure aligns conceptually with theories such as loop quantum gravity [22,23] and causal set theory [24], although QMM also emphasizes space–time quanta as active participants in information dynamics. The total Hilbert space of the QMM is constructed as the tensor product of all space–time quanta (Equation (Equation 2)):(2)$$H_{\mathrm{QMM}}=\underset{x}{⨂}H_{x},$$
meaning that the overall state of space–time is represented by a vector in $H_{\mathrm{QMM}}$ encompassing all possible quantum configurations of space–time quanta. In this discrete framework, the continuous coordinates are replaced by discrete labels, with operators corresponding to physical observables acting on these Hilbert spaces. For example, the position operators ${\hat{x}}^{\mu }$ and momentum operators ${\hat{p}}_{\nu }$ acting on $H_{x}$ satisfy the canonical commutation relations:(3)$$\left[{\hat{x}}^{\mu },{\hat{p}}_{\nu }\right]=i\hbar \delta_{\nu }^{\mu }.$$

Discretization of space–time induces quantization of geometric operators such as length, area, and volume, yielding discrete spectra [25]. For instance, the length operator $\hat{L}$ has a discrete spectrum
(4)$$\hat{L}|l_{n}\rangle =l_{n}|l_{n}\rangle ,\;l_{n}=\gamma l_{P}n,$$
where *n* is a positive integer and γ is the Barbero–Immirzi parameter [26]. Similarly, the area $\hat{A}$ and volume $\hat{V}$ operators have the following discrete spectra: (5)$$\begin{array}{cc} \hat{A}|a_{n}\rangle & =a_{n}|a_{n}\rangle ,\;a_{n}=\gamma l_{P}^{2}\sum_{i}\sqrt{j_{i}\left(j_{i}+1\right)}, \end{array}$$(6)$$\begin{array}{cc} \hat{V}|v_{n}\rangle & =v_{n}|v_{n}\rangle ,\;v_{n}=\gamma^{3/2}l_{P}^{3}\sum_{i}\sqrt{k_{i}\left(k_{i}+1\right)\left(l_{i}+1\right)}, \end{array}$$
where $j_{i}$, $k_{i}$, and $l_{i}$ are spin quantum numbers associated with the edges and nodes in the spin network representation. These operators characterize the granular geometry at the Planck scale and set a natural limit for spatial resolution, leading to intrinsic ultraviolet (UV) regularization. To ensure coordinate independence, operators and quantities in QMM are constructed as tensors or scalars under general coordinate transformations, thereby maintaining general covariance and making predictions independent of the observer’s perspective.

Figure 1 illustrates the discretized structure of space–time as hypothesized in the QMM framework. Each cell, or quantum unit, corresponds to a fundamental space–time region at the Planck scale. The axes are marked in units of $l_{P}$; fractional values (e.g., increments of $0.25\;l_{P}$) are used for illustrative purposes to enhance visual clarity, although physical lengths are quantized in integer multiples of $l_{P}$ as per Equation (Equation 4). Space–time acts not as a passive stage but as a dynamic participant, recording information via quantum imprints and bridging quantum mechanics and GR.

#### 2.1.2. Quantum Imprints

A quantum imprint is a distinctive feature of QMM, representing the recording of quantum events within space–time quanta. When a quantum field $\hat{\phi }\left(x\right)$ interacts at a space–time point *x*, the field induces a transition in the corresponding space–time quantum, encoding information about the field’s local properties.

The imprint operator ${\hat{I}}_{x}$ acting on $H_{x}$ is provided by
$${\hat{I}}_{x}=\hat{F}\left[\hat{\phi }\left(x\right),\partial_{\mu }\hat{\phi }\left(x\right),\hat{\pi }\left(x\right),\ldots \right],$$
where $\hat{F}$ depends on the field $\hat{\phi }\left(x\right)$, its derivatives $\partial_{\mu }\hat{\phi }\left(x\right)$, conjugate momenta $\hat{\pi }\left(x\right)$, and potentially higher-order terms. The form of $\hat{F}$ ensures general covariance.

For example, with a real scalar field, a possible form of $\hat{F}$ is
(7)$$\hat{F}\left[\hat{\phi }\left(x\right),\partial_{\mu }\hat{\phi }\left(x\right)\right]=g_{0}\hat{\phi }\left(x\right)+g_{1}{\hat{\phi }}^{2}\left(x\right)+g_{2}\partial_{\mu }\hat{\phi }\left(x\right)\partial^{\mu }\hat{\phi }\left(x\right),$$
where $g_{0}$, $g_{1}$, and $g_{2}$ are coupling constants. This framework allows local characteristics of quantum fields such as amplitude and energy density to be embedded in space–time itself.

#### 2.1.3. Information Conservation

Information conservation in QMM arises from the unitarity of the combined evolution of quantum fields and the QMM. The total state evolves under a Hermitian Hamiltonian $\hat{H}$, ensuring that
〈Ψ(t)|Ψ(t)〉=〈Ψ(0)|Ψ(0)〉=1,∀t,
where $|\Psi \left(t\right)\rangle \in H_{\mathrm{total}}=H_{\mathrm{fields}}⊗H_{\mathrm{QMM}}$. Thus, the QMM framework preserves the total information content, addressing unitarity in systems such as black holes where information loss paradoxes arise.

#### 2.1.4. Dynamic Interaction

Interactions between quantum fields and the QMM are governed by a local interaction Hamiltonian ${\hat{H}}_{\mathrm{int}}$:$${\hat{H}}_{\mathrm{int}}=\sum_{x}{\hat{H}}_{\mathrm{int}}\left(x\right),$$
where the local interaction at each point *x* is
(8)$${\hat{H}}_{\mathrm{int}}\left(x\right)=\hat{\phi }\left(x\right)⊗{\hat{I}}_{x}^{†}+{\hat{\phi }}^{†}\left(x\right)⊗{\hat{I}}_{x}.$$

This Hermitian form ensures energy conservation and allows for bidirectional information transfer, modifying the evolution of fields through interaction with the QMM. Unlike the holographic principle, which confines information to boundary regions, QMM envisions information distributed within the space–time volume, dynamically influencing quantum processes at each point.

#### 2.1.5. Retrieval Mechanism

QMM provides a concrete retrieval mechanism, allowing information encoded during quantum processes to influence future events. This mechanism unfolds in three phases:**Encoding Phase**: Quantum fields interacting with the QMM at an event horizon leave quantum imprints as they fall in, marking the transition from external states to encoded internal states.**Storage Phase**: The QMM retains information within space–time quanta, allowing black hole evolution while preserving information integrity.**Retrieval Phase**: During Hawking radiation emission, outgoing modes ${\hat{a}}_{k}$ interact with the QMM, transferring stored information via
(9)$${\hat{H}}_{\mathrm{retrieval}}=\sum_{k}\left(\gamma_{k}{\hat{a}}_{k}^{†}⊗{\hat{I}}_{\mathrm{HR}}+\gamma_{k}^{*}{\hat{a}}_{k}⊗{\hat{I}}_{\mathrm{HR}}^{†}\right),$$
where $\gamma_{k}$ are coupling constants and ${\hat{I}}_{\mathrm{HR}}$ represents the cumulative imprint operator, which is relevant to the Hawking radiation process. This interaction allows imprints to manifest in the radiation, suggesting that Hawking radiation may contain observable correlations from the black hole’s formation history.

The combined quantum state of the system includes both the quantum fields and the QMM residing in the total Hilbert space
(10)$$H_{\mathrm{total}}=H_{\mathrm{fields}}⊗H_{\mathrm{QMM}}.$$

The evolution of this combined state follows a modified Schrödinger equation:(11)$$i\hbar \frac{\partial }{\partial t}|\Psi \left(t\right)\rangle =\hat{H}|\Psi \left(t\right)\rangle$$
where |Ψ(t)〉 is the total state vector and $\hat{H}$ is the total Hamiltonian, which is provided by
(12)$$\hat{H}={\hat{H}}_{\mathrm{fields}}+{\hat{H}}_{\mathrm{QMM}}+{\hat{H}}_{\mathrm{int}}.$$

The components of the total Hamiltonian are:${\hat{H}}_{\mathrm{fields}}$: The Hamiltonian of the quantum fields, including kinetic and potential energy terms. For a real scalar field $\hat{\phi }\left(x\right)$, this is
(13)$${\hat{H}}_{\mathrm{fields}}=\int d^{3}x\left[\frac{1}{2}{\hat{\pi }}^{2}\left(x\right)+\frac{1}{2}{\left(\nabla \hat{\phi }\left(x\right)\right)}^{2}+V\left(\hat{\phi }\left(x\right)\right)\right],$$
where $\hat{\pi }\left(x\right)=\frac{\partial \hat{\phi }\left(x\right)}{\partial t}$ is the conjugate momentum field and $V(\hat{\phi }\left(x\right))$ represents the potential energy density.${\hat{H}}_{\mathrm{QMM}}$: This governs the intrinsic dynamics of space–time quanta, including self-interactions and neighboring interactions:
(14)$${\hat{H}}_{\mathrm{QMM}}=\sum_{x}{\hat{h}}_{x}+\sum_{\langle x,y\rangle }{\hat{h}}_{xy},$$
where ${\hat{h}}_{x}$ operates on individual quanta $H_{x}$ and ${\hat{h}}_{xy}$ represents interaction terms between neighboring quanta $H_{x}$ and $H_{y}$.${\hat{H}}_{\mathrm{int}}$: This facilitates information exchange between quantum fields and the QMM via quantum imprints, as provided in Equation (Equation 8).

Figure 2 provides a tensor network representation of space–time quanta interactions, illustrating how individual quantum cells in the QMM framework are connected. This network shows the interdependencies among quanta, reflecting the model’s approach to integrating quantum field dynamics with space–time geometry.

Unitarity is guaranteed if $\hat{H}$ is Hermitian:(15)$${\hat{H}}^{†}=\hat{H}.$$

This requires the following:${\hat{H}}_{\mathrm{fields}}^{†}={\hat{H}}_{\mathrm{fields}}$: Ensures that the field dynamics are unitary.${\hat{H}}_{\mathrm{QMM}}^{†}={\hat{H}}_{\mathrm{QMM}}$: Ensures that QMM evolution is reversible.${\hat{H}}_{\mathrm{int}}^{†}={\hat{H}}_{\mathrm{int}}$: Ensures bidirectional unitarity between the fields and QMM.

For ${\hat{H}}_{\mathrm{int}}$, this condition holds if ${\hat{\phi }}^{†}\left(x\right)=\hat{\phi }\left(x\right)$ and ${\hat{I}}_{x}^{†}={\hat{I}}_{x}$, which is the case for real scalar fields and self-adjoint imprint operators. Unitarity preservation implies that the evolution operator $\hat{U}\left(t,t_{0}\right)$ is unitary:(16)$$\hat{U}\left(t,t_{0}\right)=e^{-i\hat{H}\left(t-t_{0}\right)/\hbar },\;{\hat{U}}^{†}\hat{U}=\hat{I}.$$

Because quantum gravitational effects require regularization at Planck scales, the discretization introduces a natural UV cutoff:**Regularization**: Discrete space–time replaces integrals by sums over points:
(17)$$\int d^{4}x\to \sum_{x}\Delta V_{x}$$
where $\Delta V_{x}$ is the quantum cell’s four-volume, typically of order $l_{P}^{4}$.**Running Couplings**: Coupling constants in $\hat{F}$ and ${\hat{H}}_{\mathrm{int}}$ vary with energy scale, accounting for high-energy virtual processes; the running of coupling constants can be described by renormalization group equations, ensuring that physical predictions remain finite and well defined at all energy scales.

Considering vacuum polarization from QMM interactions, self-energy corrections to fields limited by a maximum momentum $p_{\max}∼\hbar /l_{P}$ yields the following finite integrals:(18)$$\Sigma =\int_{0}^{p_{\max}}\frac{d^{4}p}{{\left(2\pi \right)}^{4}}\frac{1}{p^{2}-m^{2}+i\epsilon }<\infty .$$

This natural UV cutoff prevents the divergences typically encountered in quantum field theories, aiding in the regularization and renormalization processes [27].

### 2.2. Clarification of Key Concepts

To ensure clarity and avoid ambiguities, we provide precise definitions and detailed explanations of key concepts within the QMM framework.

**Quantum Imprint:** A fundamental element of the QMM hypothesis, representing the record of a quantum event encoded within the state of a space–time quantum. This concept envisions a localized modification of the QMM’s state due to the interaction with a quantum field at a specific space–time point, effectively making each space–time quantum a repository of local quantum information.When a quantum field $\hat{\phi }\left(x\right)$ interacts at a space–time point *x*, it induces a change in the state of the corresponding space–time quantum. This alteration is captured by the imprint operator ${\hat{I}}_{x}$, which acts on the Hilbert space $H_{x}$ associated with that quantum cell. The imprint operator encapsulates information about the quantum field’s local properties, such as amplitude, phase, energy density, and momentum density, at that point in space–time.Quantum imprints ensure that interactions between quantum fields and the QMM are localized, with stored information that can influence future quantum events (see Figure 3). This mechanism is essential to the QMM’s capacity for information preservation, addressing the challenges of the Black Hole Information Paradox.**Space–Time Quantization:** This refers to the discretization of space–time into fundamental units at the Planck scale, resulting in a granular structure. Each space–time quantum or quantum cell occupies a finite region with dimensions on the order of the Planck length $l_{P}$ and Planck time $t_{P}=l_{P}/c$. Each quantum cell is associated with a finite-dimensional Hilbert space $H_{x}$. Physical observables localized at point *x* are represented by operators acting on $H_{x}$. This discretization is inspired by theories such as loop quantum gravity and causal set theory, which propose that space–time has an underlying discrete structure at the smallest scales [22,24].Information exchange between the quantum field ϕ(t) and the QMM through the imprinting mechanism is illustrated in Figure 4. The plot highlights the oscillatory dynamics of both the quantum field and the quantum imprint I(t), along with their respective amplitude envelopes. This figure emphasizes the reversible and dynamic nature of information transfer, consistent with the predictions of the QMM framework.**Unitarity:** A core principle of quantum mechanics asserts that the time evolution of a closed quantum system is governed by a unitary operator, preserving probability and making quantum processes reversible.In the QMM framework, unitarity is preserved by constructing the total Hamiltonian $\hat{H}$ as a Hermitian:
(19)$$\hat{H}={\hat{H}}_{\mathrm{fields}}+{\hat{H}}_{\mathrm{QMM}}+{\hat{H}}_{\mathrm{int}},\;{\hat{H}}^{†}=\hat{H}.$$Each component of the total Hamiltonian is Hermitian, ensuring the overall unitarity of the system’s evolution.**Locality and Causality:** The QMM framework respects these principles, which are crucial for consistency with quantum field theory and GR.–**Locality**: Interactions between quantum fields and the QMM occur locally at specific points in space–time. The imprint operators ${\hat{I}}_{x}$ depend on field operators and derivatives at the same point, preventing instantaneous action at a distance.–**Causality**: Information or influence does not propagate faster than light, as changes propagate within the space–time structure’s light cone. This ensures that cause precedes effect in all frames, preserving causality.**General Covariance:** A foundational principle of GR stating that physical laws are form-invariant across coordinate systems. In QMM, this principle is respected by constructing operators, interactions, and equations as scalars or tensors under general coordinate transformations:–Imprint operators ${\hat{I}}_{x}$ are formed from scalar combinations of field operators and derivatives, remaining invariant across transformations.–The interaction Hamiltonian ${\hat{H}}_{\mathrm{int}}$ is constructed as a scalar, preserving general covariance.–Evolution equations are formulated with tensorial quantities, ensuring consistency with both GR and quantum field theory.

This consistency with general covariance guarantees that predictions remain independent of the observer’s frame, which is essential for a framework aligned with GR.

In the following section, we apply the QMM framework to black hole physics, detailing information encoding during black hole formation and retrieval during Hawking radiation, then exploring the resolution to the paradox provided by QMM.

## 3. Interaction with Black Holes

### 3.1. Information Encoding During Black Hole Absorption

The QMM hypothesis provides a structured framework for understanding how information is preserved during black hole formation and evolution. In this section, we formalize the information encoding process that occurs during black hole absorption, illustrating it with mathematically detailed models. Consider a black hole formed by the collapse of matter represented by a real scalar field $\hat{\phi }\left(x\right)$. As matter collapses, intense gravitational effects near the event horizon generate strong interactions between the scalar field and the QMM. According to the QMM hypothesis, these interactions lead to quantum imprints embedded within the space–time quanta at or near the event horizon. To mathematically model this, we represent the imprint left on the QMM by an operator ${\hat{I}}_{x}$ dependent on the scalar field
(20)$${\hat{I}}_{x}=g\hat{\phi }\left(x\right),$$
where *g* is a coupling constant defining the interaction strength between the field and the QMM at point *x*. The Hamiltonian governing the interaction between the scalar field and the QMM is ${\hat{H}}_{\mathrm{int}}$, which determines the information exchange and can be expressed as
(21)$${\hat{H}}_{\mathrm{int}}=\sum_{x}\left[\hat{\phi }\left(x\right)⊗{\hat{I}}_{x}^{†}+{\hat{\phi }}^{†}\left(x\right)⊗{\hat{I}}_{x}\right].$$

For a real scalar field, where ${\hat{\phi }}^{†}\left(x\right)=\hat{\phi }\left(x\right)$ and ${\hat{I}}_{x}^{†}={\hat{I}}_{x}$, this simplifies to
(22)$${\hat{H}}_{\mathrm{int}}=2g\sum_{x}\hat{\phi }\left(x\right)⊗\hat{\phi }\left(x\right).$$

The full Hamiltonian of the combined system, including the field and QMM dynamics, is provided by
(23)$$\hat{H}={\hat{H}}_{\mathrm{fields}}+{\hat{H}}_{\mathrm{QMM}}+{\hat{H}}_{\mathrm{int}},$$
where ${\hat{H}}_{\mathrm{fields}}$ represents the Hamiltonian of the scalar field and ${\hat{H}}_{\mathrm{QMM}}$ governs the QMM dynamics as previously defined in Equation (Equation 14). The evolution of the combined state |Ψ(t)〉 in the total Hilbert space $H_{\mathrm{total}}=H_{\mathrm{fields}}⊗H_{\mathrm{QMM}}$ is governed by the Schrödinger equation:(24)$$i\hbar \frac{\partial }{\partial t}|\Psi \left(t\right)\rangle =\hat{H}|\Psi \left(t\right)\rangle .$$

To illustrate information encoding, consider an initial state where the scalar field is in a superposition of eigenstates:(25)$$|\Psi_{\mathrm{fields}}\left(0\right)\rangle =\sum_{n}c_{n}{|n\rangle }_{\mathrm{fields}},$$
where ${|n\rangle }_{\mathrm{fields}}$ are eigenstates of the field operator $\hat{\phi }\left(x\right)$ and $c_{n}$ are complex coefficients satisfying $\sum_{n}{\left|c_{n}\right|}^{2}=1$. The initial state of the QMM is taken to be $|\Psi_{\mathrm{QMM}}{\left(0\right)\rangle =|0\rangle }_{\mathrm{QMM}}$, representing a ground state with no imprints. The combined initial state is
(26)$$\left|\Psi \left(0\right)\rangle =\right|\Psi_{\mathrm{fields}}\left(0\right)\rangle ⊗|\Psi_{\mathrm{QMM}}\left(0\right)\rangle =\sum_{n}c_{n}{|n\rangle }_{\mathrm{fields}}⊗{|0\rangle }_{\mathrm{QMM}}.$$

Under the interaction Hamiltonian, the evolution leads to entanglement between the scalar field and the QMM. The state at time *t* can be formally written as follows:(27)$$|\Psi \left(t\right)\rangle =e^{-i\hat{H}t/\hbar }|\Psi \left(0\right)\rangle .$$

Expanding the evolution operator to the first order in *g* (assuming weak coupling), we have
(28)$$|\Psi \left(t\right)\rangle \approx \left(1-\frac{i}{\hbar }\hat{H}t\right)|\Psi \left(0\right)\rangle .$$

Substituting the interaction Hamiltonian, the evolved state becomes
(29)$$|\Psi \left(t\right)\rangle \approx |\Psi \left(0\right)\rangle -\frac{i}{\hbar }{\hat{H}}_{\mathrm{int}}t|\Psi \left(0\right)\rangle .$$

Applying ${\hat{H}}_{\mathrm{int}}$ to |Ψ(0)〉, we obtain
(30)$${\hat{H}}_{\mathrm{int}}|\Psi \left(0\right)\rangle =2g\sum_{n}c_{n}\sum_{x}\hat{\phi }{\left(x\right)|n\rangle }_{\mathrm{fields}}⊗\hat{\phi }\left(x\right){|0\rangle }_{\mathrm{QMM}}.$$

This indicates that the information about the coefficients $c_{n}$ becomes encoded in the QMM through the quantum imprints ${\hat{I}}_{x}=g\hat{\phi }\left(x\right)$, effectively entangling the field states with the QMM states. This encoding process preserves the information about the initial state of the collapsing matter.

Figure 5 illustrates the stages of information flow in a black hole, from encoding during matter collapse through storage as the black hole evolves to retrieval as information is emitted via Hawking radiation. This model reflects the role of the QMM in preserving information through quantum imprints in space–time, addressing the information paradox within black hole physics.

### 3.2. Hawking Radiation and Information Retrieval Mechanism

In this subsection, we explain in depth how information stored in the QMM during black hole absorption can be retrieved through Hawking radiation. We provide detailed dynamics along with numerical examples or simulations to illustrate the mechanisms wherever possible.

#### 3.2.1. Detailed Dynamics

Hawking radiation arises from quantum fluctuations near the event horizon, where particle–antiparticle pairs are created [5]. In the standard picture, one particle falls into the black hole, while the other escapes to infinity. The escaping particles form Hawking radiation, which is traditionally considered to be purely thermal and uncorrelated with the infalling matter. In the QMM framework, the outgoing Hawking radiation interacts with the QMM, allowing information stored in quantum imprints to influence the radiation. The interaction Hamiltonian between the QMM and the Hawking radiation is provided by Equation (Equation 9). The cumulative imprint operator ${\hat{I}}_{\mathrm{HR}}$ is defined as the sum over all imprints at the event horizon:(31)$${\hat{I}}_{\mathrm{HR}}=\sum_{x\in H}{\hat{I}}_{x}$$
where H denotes the set of space–time quanta at the event horizon.

The initial state of the Hawking radiation can be modeled as a vacuum state ${|0\rangle }_{\mathrm{HR}}$ with no particles. The total initial state is then
(32)$$\left|\Psi \left(0\right)\rangle =\right|\Psi_{\mathrm{fields}}\left(0\right)\rangle ⊗|\Psi_{\mathrm{QMM}}{\left(0\right)\rangle ⊗|0\rangle }_{\mathrm{HR}}.$$

Under the combined Hamiltonian $\hat{H}={\hat{H}}_{\mathrm{fields}}+{\hat{H}}_{\mathrm{QMM}}+{\hat{H}}_{\mathrm{int}}+{\hat{H}}_{\mathrm{HR}}$, the state evolves as follows:(33)$$|\Psi \left(t\right)\rangle =e^{-i\hat{H}t/\hbar }|\Psi \left(0\right)\rangle .$$

The interaction Hamiltonian ${\hat{H}}_{\mathrm{HR}}$ causes entanglement between the QMM and the Hawking radiation. This entanglement allows information stored in the QMM to be transferred to the outgoing radiation. We can express the state of the Hawking radiation after interaction as
(34)$$|\Psi_{\mathrm{HR}}\left(t\right)\rangle =\sum_{\left\{n_{k}\right\}}C_{\left\{n_{k}\right\}}\left(t\right)|\left\{n_{k}\right\}\rangle_{\mathrm{HR}}⊗{|\chi_{\left\{n_{k}\right\}}\rangle }_{\mathrm{QMM}},$$
where:$\left\{n_{k}\right\}$ represents the occupation numbers of the Hawking radiation modes.$C_{\left\{n_{k}\right\}}\left(t\right)$ are time-dependent coefficients.$|\chi_{\left\{n_{k}\right\}}\rangle_{\mathrm{QMM}}$ are states of the QMM correlated with the radiation modes.

The presence of correlations between the radiation modes and QMM states indicates that the Hawking radiation carries information about the quantum imprints, and consequently about the infalling matter.

#### 3.2.2. Concrete Example

While a full numerical simulation of this process requires extensive computational resources and a detailed model of the QMM dynamics, we can consider a simplified example. Assume that the Hawking radiation consists of a single mode *k* and that the QMM has two relevant states ${|0\rangle }_{\mathrm{QMM}}$ and ${|1\rangle }_{\mathrm{QMM}}$, respectively corresponding to the absence or presence of a quantum imprint.

The interaction Hamiltonian simplifies to
(35)$${\hat{H}}_{\mathrm{HR}}=\gamma \left({\hat{a}}_{k}^{†}⊗{\hat{\sigma }}_{+}+{\hat{a}}_{k}⊗{\hat{\sigma }}_{-}\right),$$
where ${\hat{\sigma }}_{+}={|1\rangle }_{\mathrm{QMM}}\langle 0|$ and ${\hat{\sigma }}_{-}={|0\rangle }_{\mathrm{QMM}}\langle 1|$ are the raising and lowering operators for the two-level QMM system and γ is a coupling constant.

Starting from the initial state ${|0\rangle }_{\mathrm{HR}}⊗{|1\rangle }_{\mathrm{QMM}}$, the evolution under ${\hat{H}}_{\mathrm{HR}}$ leads to
(36)$$|\Psi \left(t\right)\rangle ={\cos\left(\gamma t\right)|0\rangle }_{\mathrm{HR}}⊗{|1\rangle }_{\mathrm{QMM}}-{i\sin\left(\gamma t\right)|1\rangle }_{\mathrm{HR}}⊗{|0\rangle }_{\mathrm{QMM}}.$$

At time t=π/(2γ), the state becomes
(37)$$|\Psi \left(\frac{\pi }{2\gamma }\right){\rangle =-i|1\rangle }_{\mathrm{HR}}⊗{|0\rangle }_{\mathrm{QMM}}.$$

This indicates that the information initially stored in the QMM has been transferred to the Hawking radiation mode.

Figure 6 shows the time evolution of the quantum imprint strength during the information retrieval process. The decrease in imprint strength over time corresponds to the gradual release of information from the QMM to the Hawking radiation.

### 3.3. Implications for Black Hole Thermodynamics

The QMM hypothesis has significant implications for black hole thermodynamics, particularly concerning entropy and the laws of black hole mechanics. In classical black hole thermodynamics, the entropy $S_{\mathrm{BH}}$ of a black hole is proportional to the area *A* of its event horizon [28]:(38)$$S_{\mathrm{BH}}=\frac{k_{B}c^{3}A}{4G\hbar }.$$

This entropy is associated with the information hidden behind the event horizon. In the QMM framework, because information about the infalling matter is stored in the QMM and can be retrieved via Hawking radiation, the entropy associated with information loss decreases. The effective entropy of the black hole takes into account both the area and the information content stored in the QMM.

We can define an effective entropy $S_{\mathrm{eff}}$ as follows:(39)$$S_{\mathrm{eff}}=S_{\mathrm{BH}}-S_{\mathrm{QMM}}$$
where $S_{\mathrm{QMM}}$ represents the entropy associated with the information stored in the QMM. The second law of black hole thermodynamics states that the total entropy of a black hole and its surroundings never decreases [29]:(40)$$\Delta S_{\mathrm{total}}=\Delta S_{\mathrm{outside}}+\Delta S_{\mathrm{BH}}\geq 0.$$

With the QMM facilitating information retrieval, the entropy carried away by Hawking radiation is not purely thermal, and also includes information entropy. This modifies the entropy balance, as the information content reduces the black hole’s entropy while increasing the entropy outside due to the information-rich radiation. The **Generalized Second Law** (GSL) combines the entropy of matter and radiation outside the black hole with the black hole’s entropy [30]:(41)$$\Delta S_{\mathrm{total}}=\Delta S_{\mathrm{outside}}+\Delta S_{\mathrm{BH}}-\Delta S_{\mathrm{QMM}}\geq 0.$$

In the QMM framework, because Hawking radiation carries away information, $\Delta S_{\mathrm{outside}}$ increases not only due to thermal entropy but also due to the information entropy retrieved from the QMM. This supports the GSL while providing a mechanism that preserves unitarity. The information retrieval mechanism implies that as a black hole evaporates, information about the matter that formed it is gradually released. This affects the end stages of black hole evaporation, potentially avoiding the formation of a final singularity or remnant as the information content approaches zero.

#### Connection Between Planck Area and Volume

Black hole entropy is traditionally associated with the area of the event horizon, as in Equation (Equation 38). In the QMM framework, the information is stored in space–time quanta, each occupying a Planck volume $V_{P}=l_{P}^{3}$. The total number of quanta $N_{\mathrm{QMM}}$ involved in encoding information is related to both the area and the near-horizon volume:(42)$$N_{\mathrm{QMM}}=\frac{A\times \delta r}{l_{P}^{3}}$$
where δr is the radial thickness of the region near the event horizon, where quantum imprints are significant. Although the entropy is primarily proportional to the area, the QMM introduces a volumetric aspect to information storage, integrating both area and volume considerations.

### 3.4. Observational Signatures and Experimental Tests

If the Hawking radiation is indeed non-thermal and contains correlations reflecting the black hole’s history, this could have observable consequences:**Spectral Deviations**: Small deviations from the predicted thermal spectrum of Hawking radiation might be detectable with advanced observational techniques. The modified emission spectrum can be expressed as follows:
(43)$$\frac{dN}{dE}=\frac{\Gamma (E)}{e^{\left(E-\mu_{\mathrm{QMM}}\right)/k_{B}T_{H}}-1}$$
where Γ(E) is the greybody factor and $\mu_{\mathrm{QMM}}$ is an effective chemical potential introduced by QMM interactions. The presence of $\mu_{\mathrm{QMM}}$ implies asymmetries or spectral shifts that could be detectable. Figure 7 compares the standard Hawking radiation spectrum with the QMM-modified spectrum. The deviations from the Planckian distribution (blue curve) due to quantum imprints (red dashed curve) may be small but potentially observable.**Quantum Correlations**: Measurements of entanglement or other quantum correlations in the radiation could provide evidence for information retrieval mechanisms. Observing deviations from expected entanglement entropy, such as following the Page curve [31], would support the QMM hypothesis.**Astrophysical Observations**: Observations of black hole evaporation processes such as gamma-ray bursts from primordial black holes could offer insights. Detecting non-thermal features or unexpected correlations in the emitted radiation would indicate QMM effects.**Gravitational Wave Signatures**: QMM interactions could introduce observable corrections to gravitational wave signals from black hole mergers, particularly in the ringdown phase. Anomalies in the waveform damping or frequencies could be signatures of the QMM [32].**Cosmic Microwave Background (CMB)**: QMM-induced quantum imprints from the early universe could leave detectable signatures on the CMB, creating specific anisotropies or polarization effects that are not predicted by standard cosmological models [33].

The QMM hypothesis provides a mechanism for the encoding and retrieval of information in black hole processes. By incorporating interactions between quantum fields, the QMM, and Hawking radiation, it offers a framework that preserves unitarity and aligns with both QM and GR. In the next section, we discuss the broader implications and predictions of the QMM model, compare it with existing theories, and explore potential experimental tests that could validate or challenge the hypothesis.

## 4. Implications and Predictions of the QMM Model

In this section, we explore the profound implications of the QMM hypothesis for black hole physics, quantum gravity, and the broader understanding of fundamental physics. We provide a detailed analysis comparing the QMM model with existing theories addressing the Black Hole Information Paradox, present specific and measurable predictions arising from the QMM framework, and assess its consistency with established principles of Quantum Mechanics (QM) and General Relativity (GR). Furthermore, we anticipate potential critiques and offer responses to reinforce the viability of the QMM hypothesis.

### 4.1. Comparison with Existing Theories

The QMM hypothesis introduces a distinct perspective by positing that space–time itself acts as a dynamic quantum information reservoir capable of storing and retrieving information through quantum imprints. This approach contrasts with existing theories in several key aspects, offering unique advantages that we highlight below.

#### 4.1.1. Holographic Principle and AdS/CFT Correspondence

The holographic principle [8,9] suggests that all of the information within a volume of space can be described by degrees of freedom on its boundary. The AdS/CFT correspondence [12,15] exemplifies this principle by establishing a duality between a gravitational theory in (d+1)-dimensional anti-de Sitter (AdS) space and a conformal field theory (CFT) on its *d*-dimensional boundary.

**Comparison:** The QMM hypothesis retains the volumetric nature of space–time by introducing quantization at each point within the bulk rather than projecting information onto a lower-dimensional boundary. Unlike AdS/CFT, which relies on a specific space–time geometry, QMM operates within four-dimensional space–time, making it potentially more applicable to realistic physical scenarios.


**Unique Advantages:**
**Universality:** Independent of specific boundary conditions, allowing application across various space–time geometries.**Local Information Encoding:** Information stored locally within space–time quanta, conserving information without nonlocal mappings.**Experimental Accessibility:** By operating within observable dimensions, QMM is potentially verifiable without AdS constraints.


#### 4.1.2. Black Hole Complementarity and Firewalls

Black Hole Complementarity [34] posits that information reflects at the event horizon while also passing through it, though no observer can witness both. The **firewall paradox** [16] challenges this by suggesting an energetic “firewall” at the event horizon, violating the equivalence principle.

**Comparison:** The QMM hypothesis upholds the equivalence principle and maintains smoothness at the event horizon. It preserves information through local QMM interactions, avoiding observer-dependent realities and conflicting GR predictions.


**Unique Advantages:**
**Preservation of Fundamental Principles:** Maintains both unitarity and the equivalence principle.**Objective Information Conservation:** Information storage and retrieval are intrinsic to space–time independent of the observer.**Local Interactions:** No nonlocal effects or observer-dependent horizons are required.


#### 4.1.3. ER = EPR Conjecture

The **ER = EPR** conjecture [19] connects Einstein–Rosen (ER) bridges with quantum entanglement (EPR pairs), suggesting wormholes between entangled particles.

**Comparison:** QMM does not invoke wormholes or alter space–time topology. Instead, it quantizes space–time and stores information within its units, eliminating the need for complex topological changes.


**Unique Advantages:**
**Simpler Topology:** Operates without wormholes or topological alterations.**Planck-Scale Mechanism:** Provides insights into quantum gravity through quantization at the Planck scale.**Direct Information Embedding:** Information is embedded in space–time directly, allowing for encoding and retrieval without relying solely on entanglement.


#### 4.1.4. Soft Hair and Asymptotic Symmetries

The soft hair concept [20] posits that low-energy excitations encode information on a black hole’s horizon via asymptotic symmetries.

**Comparison:** QMM incorporates a detailed mechanism for information encoding at every space–time point through quantum imprints, expanding beyond soft hair’s scope at the horizon.


**Unique Advantages:**
**Detailed Encoding Mechanism:** Information is stored and retrieved at a fundamental level.**Applicability to All Interactions:** QMM encodes information from all quantum interactions, not only those related to soft particles or symmetries.


#### 4.1.5. Nonlocality and Quantum Gravity Effects

Some theories propose that nonlocal quantum gravity effects enable information escape from black holes [13].

**Comparison:** The QMM hypothesis preserves locality, as interactions are confined to specific points within space–time. This approach eliminates the need for nonlocal processes and maintains causal consistency.


**Unique Advantages:**
**Locality Preservation:** Simplifies theoretical models by avoiding nonlocal interactions.**Causal Consistency:** Ensures adherence to cause–effect relationships.


#### 4.1.6. Remnant Theories and Loop Quantum Gravity

Remnant theories [17] hypothesize that black hole remnants preserve information post-evaporation, but face entropy and predictability issues. Loop Quantum Gravity (LQG) [14,35] proposes quantized space–time to avoid singularities and potentially conserve information, although black hole evaporation implications remain unresolved.

**Comparison:** QMM aligns with LQG’s concept of discrete space–time, but treats quanta as active information carriers. Unlike remnant theories, QMM directly integrates information conservation within space–time itself.


**Unique Advantages:**
**Active Information Carriers:** QMM treats space–time quanta as entities storing information, not passive geometric elements.**Resolution of Black Hole Paradoxes:** QMM offers a framework for black hole information retrieval without unresolved entropy issues.


#### 4.1.7. Compatibility with Quantum Field Theory and Experimental Validation

TQMM extends quantum field theory (QFT) dynamics by enabling interactions with an underlying memory matrix that records and influences field evolution. By aligning with both QM and GR principles, QMM holds the potential for testable predictions such as observable deviations in high-energy phenomena and non-thermal correlations in Hawking radiation.


**Unique Advantages:**
**Mathematical Consistency:** Validates unitarity, locality, causality, and covariance within a consistent framework.**Experimental Accessibility:** Predictions such as Hawking radiation deviations offer pathways for testing the QMM hypothesis.


### 4.2. Observable Consequences and Testable Predictions

The QMM hypothesis leads to specific measurable predictions that could be tested through astrophysical observations or experimental setups. Below, we present these predictions along with mathematical expressions that could guide experimental efforts.

#### 4.2.1. Non-Thermal Features in Hawking Radiation

**Prediction:** Hawking radiation emitted by black holes will exhibit deviations from a perfect blackbody spectrum due to information encoded in the QMM being retrieved by the radiation.

**Mathematical Expression:** The expected particle number distribution N(ω) for Hawking radiation at frequency ω is modified from the thermal distribution $N_{\mathrm{thermal}}\left(\omega \right)$:(44)$$N\left(\omega \right)=\frac{\Gamma (\omega )}{e^{\left(\hbar \omega -\mu_{\mathrm{QMM}}\right)/k_{B}T_{H}}-1},$$
where Γ(ω) is the greybody factor, $T_{H}$ is the Hawking temperature, and $\mu_{\mathrm{QMM}}$ is an effective chemical potential introduced by QMM interactions. The presence of $\mu_{\mathrm{QMM}}$ implies asymmetries or spectral shifts that could be detectable.

**Experimental Setup:** Advanced detectors measuring the energy spectrum of radiation from black holes, such as observations of primordial black holes, could detect these deviations. Figure 7 illustrates the predicted deviations in the Hawking radiation spectrum due to QMM interactions. Detecting such deviations would provide evidence supporting the QMM hypothesis.

#### 4.2.2. Entanglement Entropy Evolution and the Page Curve

**Prediction:** The entanglement entropy $S_{\mathrm{ent}}$ of Hawking radiation over time should follow the **Page curve** [31], initially increasing and then decreasing consistent with unitary evolution.

**Mathematical Expression:** The entanglement entropy $S_{\mathrm{ent}}\left(t\right)$ is provided by
(45)$$S_{\mathrm{ent}}\left(t\right)=\left\{\begin{array}{cc} S_{\mathrm{BH}}\left(t\right), & \mathrm{for}\;t\leq t_{\mathrm{Page}},\\ S_{\mathrm{BH}}\left(t_{\mathrm{Page}}\right)-S_{\mathrm{BH}}\left(t\right), & \mathrm{for}\;t>t_{\mathrm{Page}}, \end{array}\right.$$
where $S_{\mathrm{BH}}\left(t\right)$ represents the Bekenstein–Hawking entropy of the black hole at time *t* and $t_{\mathrm{Page}}$ is the Page time when the entropy reaches its maximum.

**Experimental Setup:** While measuring $S_{\mathrm{ent}}\left(t\right)$ directly is challenging, indirect evidence could come from observing correlations in emitted radiation or through theoretical models consistent with QMM predictions.

#### 4.2.3. Modifications to Gravitational Wave Signals

**Prediction:** Gravitational waves from black hole mergers may exhibit slight deviations from classical GR predictions due to QMM interactions affecting merger dynamics.

**Mathematical Expression:** The waveform h(t) received by detectors is modified as follows:(46)$$h\left(t\right)=h_{\mathrm{GR}}\left(t\right)+\delta h_{\mathrm{QMM}}\left(t\right)$$
where $h_{\mathrm{GR}}\left(t\right)$ is the classical prediction and $\delta h_{\mathrm{QMM}}\left(t\right)$ represents QMM effects, potentially parameterized by additional terms in the inspiral or ringdown phases.

**Experimental Setup:** High-precision measurements from gravitational wave observatories such as LIGO and Virgo as well as future missions such as LISA could detect these deviations, especially in the ringdown phase.

#### 4.2.4. Quantum Interference in Black Hole Analog Systems

**Prediction:** Experiments with analog black hole systems (e.g., in Bose–Einstein condensates) will show quantum interference patterns influenced by QMM-like interactions.

**Mathematical Expression:** The interference pattern I(x) is modified as follows:(47)$$I\left(x\right)=I_{\mathrm{classical}}\left(x\right)+\delta I_{\mathrm{QMM}}\left(x\right)$$
where $\delta I_{\mathrm{QMM}}\left(x\right)$ arises from QMM-induced quantum imprints affecting the phase or amplitude of the wavefunction.

**Experimental Setup:** Laboratory experiments simulating event horizons could observe deviations in interference patterns that may indicate QMM effects using techniques in quantum optics or ultracold atoms.

#### 4.2.5. Cosmic Microwave Background (CMB) Anomalies

**Prediction:** The QMM could imprint on the CMB through quantum gravitational effects in the early universe, leading to specific anisotropies or polarization patterns not predicted by standard cosmological models.

**Mathematical Expression:** Temperature fluctuations ΔT(θ,ϕ) in the CMB may include additional terms:(48)$$\Delta T\left(\theta ,\phi \right)=\Delta T_{\mathrm{standard}}\left(\theta ,\phi \right)+\delta T_{\mathrm{QMM}}\left(\theta ,\phi \right),$$
where $\delta T_{\mathrm{QMM}}\left(\theta ,\phi \right)$ represents QMM-induced deviations potentially correlated over specific angular scales.

**Experimental Setup:** Analysis of CMB data from missions suchas Planck [33] and future missions (e.g., CMB-S4) could reveal these patterns through detailed measurements of anisotropies and polarization.

### 4.3. Consistency with Established Physics

In order for the QMM hypothesis to be a viable theory, it must be consistent with established principles of quantum mechanics and general relativity. In this subsection, we demonstrate how the QMM aligns with these fundamental frameworks and address potential critiques.

#### 4.3.1. Compatibility with Quantum Mechanics

**Unitarity and Information Conservation:** The QMM framework ensures unitarity by constructing the total Hamiltonian $\hat{H}$ to be Hermitian. The evolution of the combined system is governed by the unitary operator
(49)$$\hat{U}\left(t,t_{0}\right)=e^{-i\hat{H}\left(t-t_{0}\right)/\hbar },$$
which satisfies ${\hat{U}}^{†}\hat{U}=\hat{I}$. Information conservation is achieved, as the QMM provides a mechanism for storing and retrieving information, thereby preventing loss during processes such as black hole evaporation.

**Entanglement and Quantum Correlations:** QMM naturally incorporates entanglement between quantum fields and QMM states through interactions at each space–time point. This leads to entangled states, consistent with quantum information theory.

**No Violation of Causality or Locality:** Interactions with the QMM are local, occurring at specific space–time points, and propagate according to the standard causal structure of space–time, maintaining consistency with the principle of locality in quantum mechanics.

#### 4.3.2. Compatibility with General Relativity

**Equivalence Principle Preservation:** QMM does not alter the smoothness of space–time at macroscopic scales or the experience of an infalling observer crossing the event horizon, preserving the equivalence principle.

**Recovery of Classical GR in the Appropriate Limit:** At scales much larger than the Planck length, the effects of space–time quantization become negligible and the QMM framework reduces to classical GR, ensuring the validity of well-tested GR predictions.

**Energy–Momentum Conservation:** interactions between quantum fields and the QMM are designed to conserve energy and momentum. The total energy–momentum tensor includes contributions from both the fields and the QMM, satisfying the conservation law:(50)$$\nabla_{\mu }T_{\mathrm{total}}^{\mu \nu }=0.$$

#### 4.3.3. Addressing Potential Critiques


**Critique: Complexity and Testability**
–*Objection:* The QMM introduces additional complexity to the theoretical framework, and its predictions may be challenging to test experimentally.–*Response:* While the QMM adds complexity at the Planck scale, it provides a concrete mechanism for resolving the Black Hole Information Paradox without violating established principles. Predictions such as deviations in Hawking radiation and gravitational wave signals offer pathways for experimental verification as technology advances.
**Critique: Integration with Quantum Gravity Theories**
–*Objection:* The QMM needs to be reconciled with existing quantum gravity theories such as loop quantum gravity or string theory.–*Response:* The QMM shares common ground with loop quantum gravity in terms of space–time quantization and can be viewed as a complementary approach that focuses on information storage within space–time. Further research could explore how the QMM integrates with or enhances existing quantum gravity frameworks.
**Critique: Renormalization and UV Behavior**
–*Objection:* The behavior of the QMM at high energies and issues related to renormalization need to be addressed to ensure consistency.–*Response:* The discretization of space–time at the Planck scale provides a natural ultraviolet (UV) cutoff, potentially aiding in regulating divergences. Standard renormalization techniques can be adapted within the QMM framework to handle high-energy behavior, ensuring mathematical consistency. For example, self-energy corrections to fields are limited by the maximum momentum $p_{\max}∼\hbar /l_{P}$, yielding finite integrals, as shown in Equation (Equation 18).
**Critique: Lack of Direct Evidence**
–*Objection:* There is currently no direct experimental evidence supporting the existence of the QMM.–*Response:* The QMM makes specific predictions that could be tested with future advancements in observational technology. The absence of direct evidence is common in theories addressing phenomena at the Planck scale. The consistency of QMM with established physics and its potential to resolve longstanding paradoxes justify its consideration.

### 4.4. Advantages Over Existing Theories

The QMM hypothesis offers a framework that addresses the Black Hole Information Paradox by proposing a mechanism for information storage and retrieval within the quantized fabric of space–time. Distinct from theories relying on boundary projections or exotic constructs, QMM encodes information locally through quantum imprints at each point in space–time, preserving unitarity and causality in alignment with both quantum mechanics and general relativity.

QMM’s advantages over existing theories include:**Preservation of Fundamental Physical Principles:** QMM maintains both unitarity and the equivalence principle without invoking exotic phenomena such as firewalls or wormholes.**Local Interactions and Causal Consistency:** All information storage and retrieval interactions are local and respect the causal structure of space–time.**Applicability to Observable Universe:** QMM operates within four-dimensional space–time and does not require additional dimensions or specific geometries, making it compatible with our observable universe.**Potential for Experimental Verification:** QMM predicts measurable deviations in known physical processes, such as non-thermal features in Hawking radiation and gravitational wave anomalies, opening pathways for experimental validation.

In the subsequent section, we discuss potential experimental setups and observational strategies for testing the predictions of the QMM hypothesis, aiming to bridge the gap between theoretical constructs and empirical evidence.

## 5. Experimental Implications and Testable Predictions

The QMM hypothesis, while theoretical, suggests various experimental implications and testable predictions that can guide empirical validation. This section explores the observable effects predicted by the QMM model, evaluates the feasibility of their detection with existing or emerging technologies, suggests specific experimental setups and observations, and highlights opportunities for interdisciplinary collaboration.

### 5.1. Potential Observable Effects

The QMM model proposes several phenomena that would substantiate the hypothesis if observed. Key predicted effects include deviations in Hawking radiation spectra, alterations in entanglement entropy evolution, modifications to gravitational wave signals, anomalies in cosmic microwave background (CMB) observations, and observable phenomena in laboratory analogs.

#### 5.1.1. Non-Thermal Features in Hawking Radiation

The QMM framework posits that Hawking radiation emitted by black holes is not purely thermal but carries information through quantum imprints, causing deviations from a perfect blackbody spectrum [5]. This occurs as the radiation interacts with the QMM, transferring information about infalling matter and modifying the statistical properties of the emitted spectrum. The resulting modified emission spectrum can be expressed as Equation (Equation 43) where:$\frac{dN}{dE}$ is the particle number per unit energy interval (particles/eV).Γ(E) represents the greybody factor accounting for potential barriers around the black hole.$\mu_{\mathrm{QMM}}$ is an effective chemical potential introduced by QMM interactions (eV).$k_{B}$ is Boltzmann’s constant ($8.617\times 10^{-5}$ eV/K).$T_{H}$ is the Hawking temperature (K).*E* is the particle energy (eV).

The presence of $\mu_{\mathrm{QMM}}$ implies asymmetries or spectral shifts that could be detectable in observed Hawking radiation (see Figure 7). Figure 7 illustrates the modified energy spectrum of Hawking radiation when influenced by the QMM. The spectrum reflects deviations from a pure blackbody radiation pattern due to quantum imprints, suggesting detectable differences in emitted radiation that could validate the QMM hypothesis through empirical observation.

#### 5.1.2. Entanglement Entropy Evolution and the Page Curve

The QMM model anticipates that the entanglement entropy $S_{\mathrm{ent}}$ of Hawking radiation will trace the **Page curve**, a characteristic trajectory representing unitary evolution [31]. Initially, $S_{\mathrm{ent}}$ should increase as the black hole radiates, reach a maximum at the Page time $t_{\mathrm{Page}}$, and then decrease as the black hole fully evaporates, indicating that information returns to the environment. Mathematically, this entanglement entropy behavior can be modeled as
(51)$$S_{\mathrm{ent}}\left(t\right)=\min\left[S_{\mathrm{BH}}\left(t\right),S_{\mathrm{BH}}^{\mathrm{initial}}-S_{\mathrm{BH}}\left(t\right)\right],$$
where:$S_{\mathrm{BH}}\left(t\right)=\frac{k_{B}c^{3}A\left(t\right)}{4G\hbar }$ is the Bekenstein-Hawking entropy at time *t* (J/K).$S_{\mathrm{BH}}^{\mathrm{initial}}$ represents the black hole’s initial entropy (J/K).A(t) is the area of the event horizon at time *t* (m^2^).$k_{B}$ is Boltzmann’s constant.*c* is the speed of light in vacuum ($3.00\times 10^{8}$ m/s).*G* is the gravitational constant ($6.674\times 10^{-11}$ m^3^ kg^−1^ s^−2^).*ℏ* is the reduced Planck constant ($1.055\times 10^{-34}$ J s).

Thus, the entropy of the radiation mirrors the black hole’s entropy reduction after $t_{\mathrm{Page}}$, implying that information is conserved and gradually released, consistent with QMM’s information retrieval mechanism.

#### 5.1.3. Modifications to Gravitational Wave Signals

The QMM framework suggests that quantum field interactions with the QMM could introduce observable corrections to gravitational wave signals from black hole mergers. This would be particularly noticeable in the waveform during the ringdown phase, where additional damping might occur [32]. The detected gravitational wave strain h(t) can be expressed as
(52)$$h\left(t\right)=h_{\mathrm{GR}}\left(t\right)+\delta h_{\mathrm{QMM}}\left(t\right),$$
where:$h_{\mathrm{GR}}\left(t\right)$ is the classical GR prediction of the strain.$\delta h_{\mathrm{QMM}}\left(t\right)$ denotes the correction induced by QMM interactions.

This correction can be modeled as
(53)$$\delta h_{\mathrm{QMM}}\left(t\right)=\epsilon \;h_{\mathrm{GR}}\left(t\right)\;e^{-\kappa t},$$
where:ϵ is a small dimensionless parameter characterizing QMM interaction strength.κ is a damping coefficient (s^−1^).*t* is time since the merger (s).

Detecting such deviations would require precise measurements of gravitational waves, particularly during the ringdown phase, where the QMM effects may be most pronounced.

#### 5.1.4. Anomalies in Primordial Black Hole Evaporation

For primordial black holes (PBHs) potentially evaporating in the present epoch, the QMM model predicts that quantum imprints could influence the emission spectrum, producing detectable cosmic rays or gamma rays with unique non-thermal features [36]. Anomalies in energy spectra or unexpected correlations in detected cosmic ray flux would point towards QMM interactions.

#### 5.1.5. Cosmic Microwave Background (CMB) Signatures

QMM-induced quantum imprints from the early universe could leave detectable signatures on the CMB, creating specific anisotropies or polarization effects not predicted by standard cosmological models [33]. The temperature fluctuations ΔT(θ,ϕ) could include QMM-induced deviations as follows:(54)$$\Delta T\left(\theta ,\phi \right)=\Delta T_{\mathrm{standard}}\left(\theta ,\phi \right)+\delta T_{\mathrm{QMM}}\left(\theta ,\phi \right),$$
where:$\Delta T_{\mathrm{standard}}\left(\theta ,\phi \right)$ represents the temperature fluctuations predicted by the ΛCDM model (K).$\delta T_{\mathrm{QMM}}\left(\theta ,\phi \right)$ represents potential contributions from QMM effects (K).θ and ϕ are angular coordinates on the sky (degrees).

Statistical analysis of CMB data may reveal these deviations, providing insights into QMM’s role in the early universe.

#### 5.1.6. Laboratory Analog Gravity Experiments

Experimental setups with Bose–Einstein condensates (BECs), optical lattices, or superconducting circuits may exhibit phenomena analogous to QMM quantum imprints and information retrieval [37]. Observing non-thermal correlations or emissions in these controlled systems would indirectly support the QMM model. For example, analog black hole horizons created in BECs can simulate Hawking radiation, where deviations from expected thermal spectra could indicate QMM-like effects.

### 5.2. Feasibility of Detection

The feasibility of detecting these QMM effects relies on both current capabilities and future advancements in observational technology and laboratory techniques.

#### 5.2.1. Technological Requirements

High-sensitivity gamma-ray observatories are essential for detecting deviations in Hawking radiation or PBH signatures. Instruments such as the Cherenkov Telescope Array (CTA) [38] and the Fermi Gamma-ray Space Telescope [39] are capable of detecting high-energy gamma rays potentially emitted by evaporating PBHs. For gravitational wave analysis, detectors such as LIGO [40], Virgo [41], KAGRA [42], and future systems such as LISA (Laser Interferometer Space Antenna) [43] offer enhanced sensitivity in detecting potential QMM effects. Advances in CMB research, particularly with missions such as Planck [33], LiteBIRD [44], or ground-based telescopes such as the Atacama Cosmology Telescope (ACT) [45], could support searches for early-universe QMM-induced anomalies. Finally, quantum simulation platforms, including ultra-cold atom setups [46], optical lattices, and superconducting qubit arrays [47], could enable simulation of QMM interactions and space–time discretization.

#### 5.2.2. Current Capabilities vs. Future Advancements

Current technologies such as gravitational wave observatories have provided data on black hole and neutron star mergers, offering preliminary insights for detecting waveform anomalies. CMB data from Planck provide high-precision measurements conducive to anomaly detection. Laboratory analogs in BECs and optical systems could simulate black hole phenomena, and operational quantum simulators could allow for basic QMM modeling. Future gravitational wave detectors with improved sensitivity, enhanced CMB measurement precision, advancements in quantum computing, and ultra-sensitive detectors will all contribute to more accurate and direct QMM hypothesis testing. For instance, next-generation gravitational wave observatories such as the Einstein Telescope [48] and Cosmic Explorer [49] could detect subtler effects predicted by QMM.

### 5.3. Suggested Experimental Setups

Several targeted experimental and observational approaches could help t test the predictions of the QMM hypothesis.

**Monitoring for Primordial Black Hole Evaporation:** Gamma-ray observatories such as CTA and Fermi could search for high-energy bursts consistent with PBH evaporation. By analyzing energy spectra and temporal profiles for deviations predicted by QMM interactions, observed non-thermal spectral features or temporal variations that diverge from conventional astrophysical models would suggest QMM effects.**Gravitational Wave Signal Analysis:** Utilizing gravitational wave data from LIGO, Virgo, and KAGRA, scientists can analyze waveforms for anomalies potentially arising from QMM interactions, particularly during the ringdown phase. Comparing detected signals with theoretical models incorporating QMM corrections could reveal discrepancies from classical GR predictions, providing evidence for QMM effects.**CMB Anisotropy and Polarization Studies:** Data from *Planck*, ACT, the *South Pole Telescope* (SPT) [50], and forthcoming missions such as *LiteBIRD* and CMB-S4 [51] could be analyzed for statistical anomalies aligned with QMM-induced fluctuations. Advanced statistical techniques and machine learning can enhance the sensitivity of searches for deviations from the ΛCDM model.**Laboratory Analog Gravity Experiments:** Experiments using BECs to create acoustic black holes may simulate Hawking radiation and enable observation of information retention and retrieval analogous to QMM processes [37]. Optical systems with engineered refractive indices can mimic event horizons, facilitating study of photon dynamics similar to those predicted by the QMM model [52]. Additionally, superconducting circuits could mimic space–time discretization, allowing for direct observation of QMM-like quantum information transfer [53].**Quantum Simulation and Computation:** Quantum simulators and computers provide a platform for modeling interactions between quantum fields and a discretized space–time QMM. Algorithms designed to simulate the modified evolution of quantum states under QMM interactions would provide valuable insights into the theoretical aspects of the QMM hypothesis [54].**Advanced Quantum Sensors and Interferometry:** Ultra-sensitive interferometers or quantum sensors could measure fluctuations in space–time metrics that may be attributable to QMM interactions, potentially by utilizing atom interferometry [55] or optomechanical systems [56]. Directly detecting quantum imprint signatures or quantization effects in space–time would significantly bolster the QMM hypothesis.

### 5.4. Opportunities for Interdisciplinary Collaboration

The QMM hypothesis offers numerous potential avenues for experimental validation necessitating collaboration across disciplines:**Astrophysics and Cosmology:** Collaboration between theoretical physicists and astronomers is essential for designing observations to detect QMM effects in cosmic phenomena such as black hole evaporation and CMB anomalies.**Gravitational Wave Astronomy:** Joint efforts between gravitational wave observatories and theorists could focus on refining waveform models to include QMM corrections and developing data analysis techniques sensitive to these effects.**Quantum Optics and Condensed Matter Physics:** Laboratory experiments simulating black hole analogs require expertise in quantum optics, BECs, and superconducting systems in order to model and detect QMM-like phenomena.**Quantum Information Science:** Developing quantum simulation algorithms and leveraging quantum computing resources involve collaboration with computer scientists and quantum information theorists.**High-Energy Physics:** Understanding the implications of QMM on particle interactions at high energies can benefit from experimental input from particle accelerators and detectors.

These experimental and observational approaches highlight the intersection of astrophysics, quantum mechanics, and laboratory physics, paving the way for a deeper understanding of quantum gravity and the fundamental nature of space–time.

## 6. Conclusions

In this paper, we have introduced the Quantum Memory Matrix (QMM) hypothesis, proposing that space–time itself functions as an active quantum information repository through the mechanism of quantum imprints. These quantum imprints represent localized modifications in space–time quanta, encoding information about quantum events at each point in space–time. By incorporating space–time quantization at the Planck scale, the QMM framework provides a mathematically consistent model for information preservation and retrieval, ensuring unitarity in black hole scenarios. Unlike traditional approaches such as the holographic principle or black hole complementarity, which rely on information storage on boundary surfaces or involve observer-dependent realities, the QMM hypothesis embeds information within the volumetric structure of space–time. This approach allows information to be preserved and dynamically interact with the evolution of quantum fields without modifying classical general relativity (GR) predictions at macroscopic scales. Through a detailed mathematical framework, we have derived interaction Hamiltonians and demonstrated the preservation of unitarity, locality, and causality within the QMM framework. The total Hamiltonian of the combined system, including quantum fields and the QMM, is constructed to be Hermitian, ensuring that the time evolution is unitary. We show that interactions between quantum fields and the QMM are local and conserve energy–momentum, satisfying the conservation laws consistently with both quantum mechanics and general relativity. The QMM hypothesis provides potential explanations for the Black Hole Information Paradox by offering a mechanism for information storage and retrieval within the quantized fabric of space–time. The QMM model predicts specific measurable deviations in physical phenomena, such as modifications of the Hawking radiation spectrum and gravitational wave signals. These predictions open pathways for experimental exploration which could lead to advancements in understanding the interplay between quantum information dynamics and the structure of space–time.

Further research could involve developing explicit models for quantum imprints in various quantum field theories, performing numerical simulations of black hole processes within the QMM framework, and designing experiments to search for QMM-induced effects in astrophysical observations and laboratory analogs. Investigating connections between the QMM hypothesis and existing quantum gravity theories may also provide deeper insights into the unification of QM and GR.

The QMM hypothesis offers a framework that aligns with established physical principles and provides a potential solution to longstanding problems in theoretical physics. By embedding information within space–time itself, the hope for the QMM model is to contribute to the broader understanding of the fundamental nature of space–time and quantum information. 

## Figures and Tables

**Figure 1 entropy-26-01039-f001:**
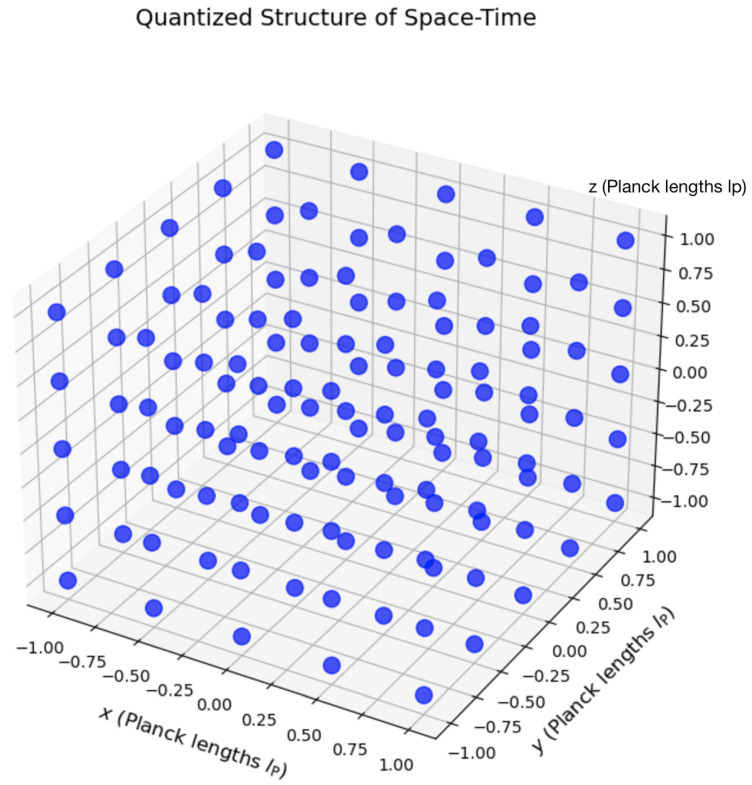
Visualization of space–time quantization. Each cell represents a discrete quantum unit in the QMM framework, showing the granularity at the Planck scale. The axes are labeled in units of Planck length $l_{P}$.

**Figure 2 entropy-26-01039-f002:**
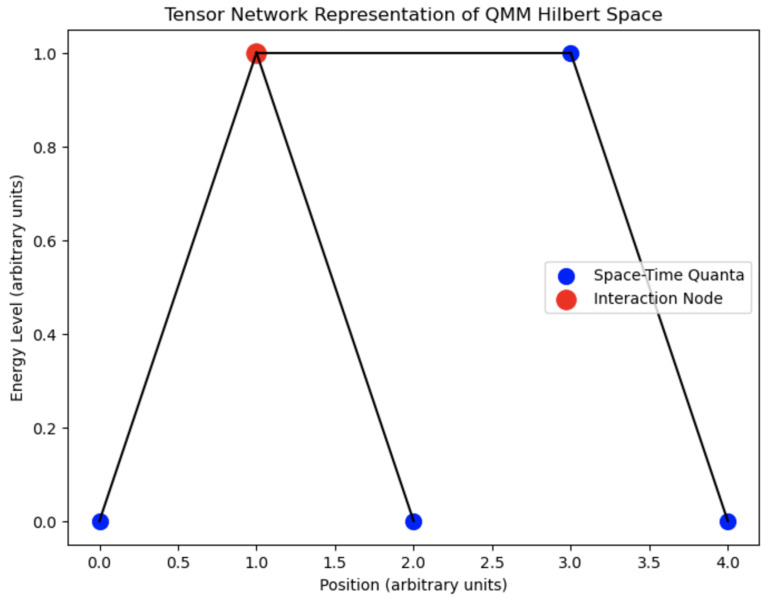
Tensor network structure representing interactions among space–time quanta in the QMM; nodes represent quantum cells and edges represent interactions between neighboring cells.

**Figure 3 entropy-26-01039-f003:**
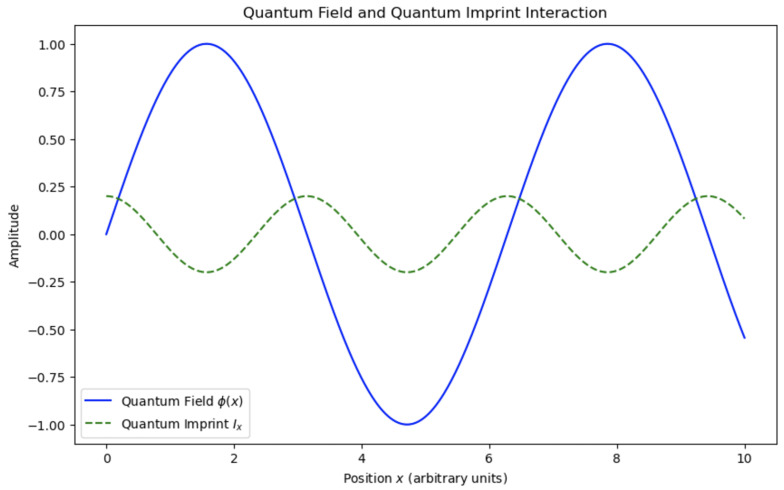
Interaction between a quantum field and a quantum imprint, showing how a quantum event leaves an imprint in the QMM’s state. The units on the axes are in Planck length $l_{P}$ and Planck time $t_{P}$.

**Figure 4 entropy-26-01039-f004:**
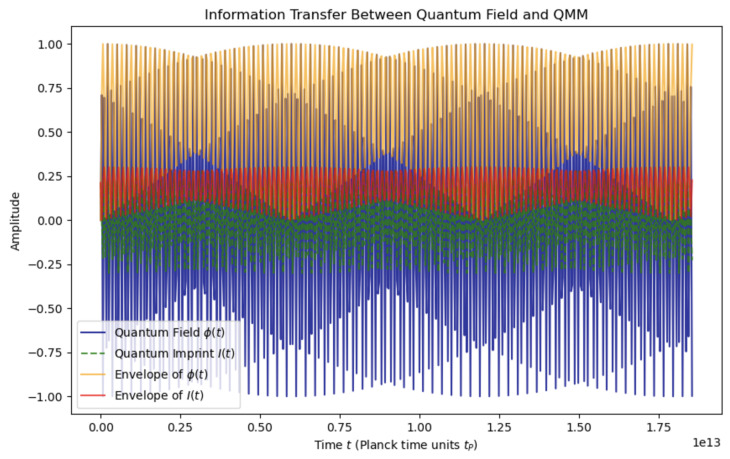
Information exchange between the quantum field ϕ(t) and the QMM through the imprinting mechanism as modeled in the QMM framework. The time axis is expressed in units of Planck time ($t_{P}$), emphasizing the quantum scale of the interaction.

**Figure 5 entropy-26-01039-f005:**
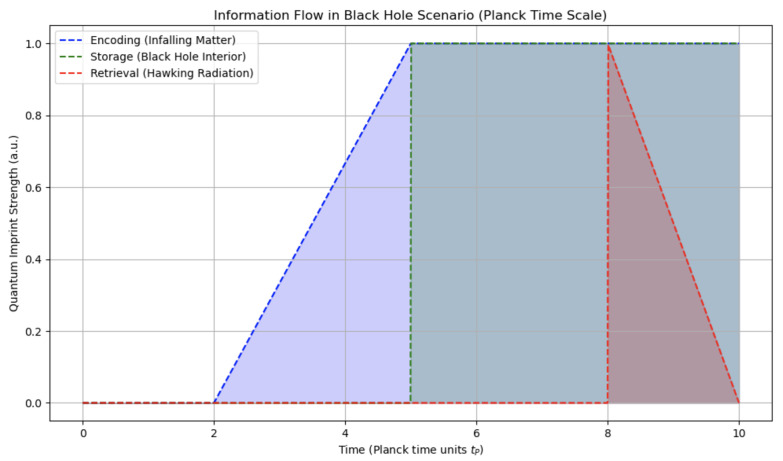
Information flow in a black hole scenario according to the QMM hypothesis. The encoding phase captures information from infalling matter at the event horizon (blue arrows). During the storage phase, information remains within the QMM (green region). In the retrieval phase, information is gradually emitted as Hawking radiation (red arrows). Units on the spatial axes are in kilometers (km). Time is in seconds (s), corresponding to macroscopic black hole scales.

**Figure 6 entropy-26-01039-f006:**
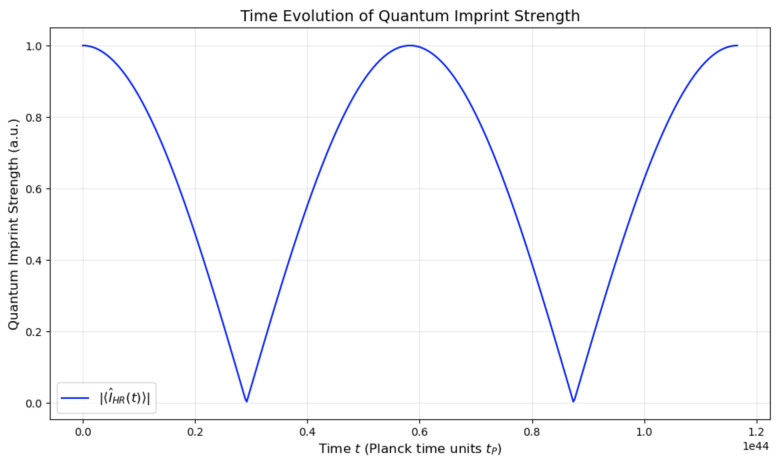
Time evolution of the quantum imprint strength during the information retrieval process. The quantum imprint strength $\left|\langle {\hat{I}}_{\mathrm{HR}}\left(t\right)\rangle \right|$ evolves under the interaction Hamiltonian ${\hat{H}}_{\mathrm{HR}}$ described in Equation (Equation 35), which governs the transfer of information between the QMM and Hawking radiation. The vertical axis represents the quantum imprint strength in arbitrary units (a.u.), and the horizontal axis represents time in seconds (s). The oscillatory behavior is given by $\left|\langle {\hat{I}}_{\mathrm{HR}}\left(t\right)\rangle |=|\cos\left(\gamma t\right)\right|$, where γ is the interaction coupling constant (set to γ=1.0 in arbitrary units for this simulation). The periodic decrease to zero in imprint strength corresponds to the moment when information from the QMM is fully transferred to the Hawking radiation mode. The simulation is performed over a time range of t∈[0,2π/γ] using 500 data points for resolution. All numerical values in the figure are expressed in scientific notation, e.g., $8\times 10^{3}$ instead of 8E3.

**Figure 7 entropy-26-01039-f007:**
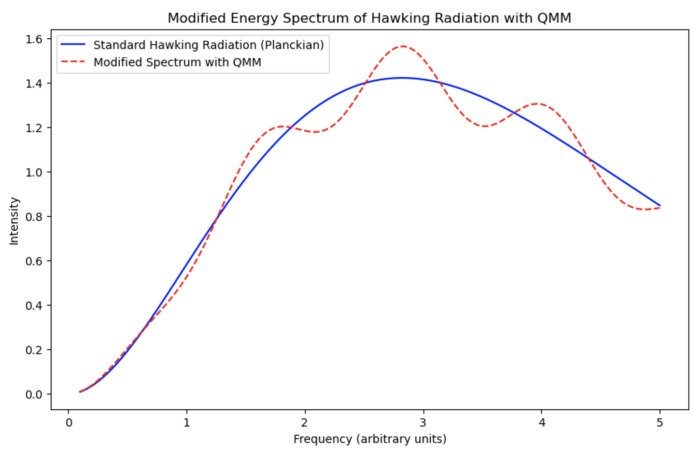
Modified energy spectrum of Hawking radiation as predicted by the QMM framework. The horizontal axis represents energy *E* in electronvolts (eV), while the vertical axis represents the particle number per unit energy interval $\frac{dN}{dE}$ in particles per electronvolt (particles/eV). The standard Hawking spectrum (solid blue line) is modeled as a Planckian distribution $I\left(E\right)\propto \frac{E^{3}}{e^{E/k_{B}T}-1}$, where $k_{B}$ is Boltzmann’s constant and *T* is the Hawking temperature. The QMM-modified spectrum (dashed red line) introduces deviations through an oscillatory modulation, reflecting the influence of quantum imprints on the emitted radiation. The modified spectrum is modeled as $I_{\mathrm{QMM}}\left(E\right)=I_{\mathrm{standard}}\left(E\right)\cdot \left(1+\alpha \sin(\beta E)\right)$, where α=0.1 and β=5.0 are parameters controlling the amplitude and frequency of the imprints. The figure demonstrates how the QMM affects the high-energy regime, suggesting measurable deviations from the standard spectrum.

## Data Availability

No new data were created or analyzed in this study. All theoretical results are derived from the proposed Quantum Memory Matrix framework, and no datasets are associated with this research.

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
