# Peer review of "The Quantum Memory Matrix: A Unified Framework for the Black Hole Information Paradox"

_entropy, 2024, doi:10.3390/e26121039_

Round 1
Reviewer 1 Report
Comments and Suggestions for Authors
There are 78 references listed in the References section at the end, but less than half of these appear in the body of the paper. Further the references that do appear are out of order -- the first reference listed on page 1 is [10] and there is no order after this. And again over half the references in the reference section do not appear in the body of the paper.
Also many parts of the paper appear to be simply a listing of results/statements with little or no context -- see many bullet points on pages 14-18. This just seems to be a shopping list of terms.
On page 3 eqns. 3 and 4 give the discrete spectra for the area and volume operators. What about the discrete spectrum for position/length operator? Just above eqn. (3) the authors mention discretization of space-time but then only list the discrete area and volume. What about length?
What are the units for x, y and z axes in figure 1? Are these Planck length? If so does the fact that these go up in increments of 0.25 mean fractional Planck length are considered?
For figure 2 what are the units for the horizontal position axis? what are the units of the vertical energy level axis?
On page 7 the authors say "Running Couplings: Coupling constants in ˆF and ˆH_int vary with energy scale, accounting for high-energy virtual processes.
Consider vacuum polarization from QMM interactions. Self-energy corrections to fields, limited by p_max ∼ h/l_P, yield finite integrals."
How do the couplings run with energy scale? Do they get larger or smaller with increasing energy scale? Also can the authors show explicitly with a concrete calculation that "Self energy corrections.... yield finite integrals."
For figure 5 the vertical axis is labeled "Quantum Imprint Strength". Where is the formula for this and what are the units? Also for the horizontal time axis what are the units? Seconds? Planck time?
On page 18 the authors claim "The discretization of space-time at the Planck scale provides a natural UV cutoff, potentially aiding in regulating divergences. Standard renormalization techniques can be adapted within the QMM framework to handle high-energy behavior, ensuring mathematical consistency." If they can do this then this big news. However, I am very skeptical. Can they show with concrete calculations that this is in fact the case.
In figure 6 the authors present the Hawking radiation spectrum -- intensity versus frequency. The standard result is the blue curve and the QMM result is the red, dashed curve. However, the standard Hawking radiation spectrum is a Planckian spectrum and the blue curve does not look Planckian. And again not units are given for either intensity or frequency. Finally, what formula is used to generate the modified spectrum i.e. the red dashed curve?
Section 5.3 lists a series of experimental set-ups for testing QMM. However no references to any of these experimental set-ups is given. The authors need to provide references for each bullet point on pages 20-21
Comments on the Quality of English Language
The English is fine.
Reviewer 2 Report
Comments and Suggestions for Authors
This paper proposes a solution of the black hole information loss problem by the Quantum Memory Matrix (QMM) approach. QMM is the reservoir of information made of spacetime blocks each of the Planck volume (Vp = lp3 ) which encodes information hidden by a black hole while in the process of a collapse. This information is then stored and later released during the evaporation process according to the Page curve behavior. There is a process of interaction between the QMM and the quantum fields simulated by the appropriate interaction term in the Hamiltonian (14).
An idea is new (though there have been some proposals of this type which included entanglement) and worth publishing, but I have some fundamental questions which should be clarified in the paper.
They are as follows:
- QMM considers Planck volume (Vp = lp3 ) - see Fig. 1 – as the fundamental bit of information, but one usually expresses black hole entropy in terms of Planck area (Ap = lp2 ). In the paragraph by Eq. (16) (lines 282-283) the Authors say “spacetime quanta at or near the event horizon”. Here “at” for me means the area while “near” means volume. How does it combine both? Or perhaps horizon is not considered area here?
2. In relation to the above – the formula (36) includes the area only. How about the QMM bit which is volume?
3. Small point: Eq. (19) is repeated after Eq. (14).
